# Surface Modulation of 3D Porous CoNiP Nanoarrays In Situ Grown on Nickel Foams for Robust Overall Water Splitting

**DOI:** 10.3390/ijms23105290

**Published:** 2022-05-10

**Authors:** Jianpeng Li, Caiyan Gao, Haiyang Wang, Baojun Li, Shufang Zhao, Young Dok Kim, Zhongyi Liu, Xin Du, Zhikun Peng

**Affiliations:** 1College of Chemistry, Research Center of Green Catalysis, Henan Institute of Advance Technology, Zhengzhou University, Zhengzhou 450001, China; lijianpeng@gs.zzu.edu.cn (J.L.); 18037897076@163.com (H.W.); lbjfcl@zzu.edu.cn (B.L.); zhongyiliu@zzu.edu.cn (Z.L.); pengzhikun@zzu.edu.cn (Z.P.); 2School of Resources and Environmental Engineering, Wuhan University of Technology, Wuhan 430070, China; 17803897978@163.com; 3Department of Chemistry, Sungkyunkwan University, Suwon 16419, Korea; taylorbjt096-@naver.com (S.Z.); ydkim91@skku.edu (Y.D.K.)

**Keywords:** CoNiP NA/NF, porous nanoarrays, bifunctional catalyst, overall water splitting

## Abstract

The careful design of nanostructures and multi-compositions of non-noble metal-based electrocatalysts for highly efficient electrocatalytic hydrogen and oxygen evolution reaction (HER and OER) is of great significance to realize sustainable hydrogen release. Herein, bifunctional electrocatalysts of the three-dimensional (3D) cobalt-nickel phosphide nanoarray in situ grown on nickel foams (CoNiP NA/NF) were synthesized through a facile hydrothermal method followed by phosphorization. Due to the unique self-template nanoarray structure and tunable multicomponent system, the CoNiP NA/NF samples present exceptional activity and durability for HER and OER. The optimized sample of CoNiP NA/NF-2 afforded a current density of 10 mA cm^−2^ at a low overpotential of 162 mV for HER and 499 mV for OER, corresponding with low Tafel slopes of 114.3 and 79.5 mV dec^−1^, respectively. Density functional theory (DFT) calculations demonstrate that modulation active sites with appropriate electronic properties facilitate the interaction between the catalyst surface and intermediates, especially for the adsorption of absorbed H* and *OOH intermediates, resulting in an optimized energy barrier for HER and OER. The 3D nanoarray structure, with a large specific surface area and abundant ion channels, can enrich the electroactive sites and enhance mass transmission. This work provides novel strategies and insights for the design of robust non-precious metal catalysts.

## 1. Introduction

The excessive consumption of fossil fuels has resulted in serious environmental pollution and energy crisis. Green and sustainable energy needs to be developed urgently for sustainable economic development [1]. Hydrogen, with the characterizations of high calorific value and being pollution-free, is an ideal substitute for fossil fuels [2,3,4]. Electrocatalytic water splitting is a green technology for hydrogen production that converts renewable energy (such as solar energy and wind energy) into hydrogen energy, which has a wide application prospect. The theoretical potential to drive water splitting is 1.23 V. At present, a significant challenge is the development of highly efficient, stable, and inexpensive catalysts for hydrogen evolution reaction (HER) and oxygen evolution reaction (OER) to lower the high energy consumption caused by large overpotentials. Noble metals (such as Pt and Ru) and their oxides (such as IrO_2_ and RuO_2_) are recognized as the current benchmarking HER and OER catalysts, respectively. However, their high cost and low reserves severely limit the practical applications [5,6]. Therefore, researchers are committed to develop high-efficiency and earth-abundant alternatives.

Non-noble metal, including its hydroxides, chalcogenides, carbides, phosphides and nitrides have been reported as promising HER catalysts due to unique electronic properties and superior conductivity [7,8,9,10,11]. Among these materials, transition metal phosphides (TMPs) have attracted a great deal of attention owing to their unique electronic properties and great conductivity [12,13,14]. However, monometallic phosphides, limited by their inherent properties, have difficulty achieving the desired catalytic performance for HER/OER. Multicomponent metal phosphides can realize the optimization of surface/interface electronic structures and benefit from the synergistic effects to improve the HER and/or OER performance. Wang et al. constructed a Co_3_P/Ni_2_P heterojunction, achieving 10 mA cm^−2^ at an overpotential of 115 mV for HER [15]. The strong interactions at the heterojunction interface can optimize the electronic environment of the catalytic surface, leading to suitable H^+^ adsorption and hydrogen formation kinetics. Wang and colleagues reported the phase-controlled synthesis of iron/nickel phosphide nanocrystals “armored” with porous P-doped carbon. Using (Ni_x_Fe_1−x_)_2_P@PC/PG as cathode and anode catalysts, it needs only low cell voltage (1.45 V) to reach a 10 mA cm^−2^ current [16]. These works confirm that the fabrication of TMPs with multi-metal components can mediate the electronic structure of TMPs, thereby providing more appropriate activity sites for HER/OER. A series of strategies have been explored, such as doping, bimetallic phosphide synthesis, heterostructure construction, and interfacial engineering [15]. The systemic modulation of the electronic properties of active sites of bimetallic phosphating alloys in combination with the composition and micromorphology is still challenging. The catalytic performance of the catalyst is not only related to the intrinsic properties of active sites, but is also connected with the morphology. The monolithic integrated nanoarray catalysts carry a more stable structure, a larger surface area for more active site exposure, and a porous structure, and thus a more favorable mass transfer processes can be carried out, thereby manifesting enhanced catalytic performance [17,18,19].

Here, we have systematically studied bimetallic phosphating alloys with tunable composition and morphology. By adjusting the ratio of Co/Ni metal precursors, a series of CoNiP nanoarrays with different micromorphology and surface properties were synthesized. Among them, the optimal CoNiP NA/NF-2 exhibited excellent activity and stability for both HER and OER. It afforded the current density of 10 mA cm^−2^ at a low overpotential of 162 mV for HER and 499 mV for OER, corresponding with low Tafel slopes of 114.3 and 79.5 mV dec^−^^1^, respectively, as well as robust durability. It required a cell potential of only 1.613 V to achieve the current density of 10 mA cm^−2^. The superior HER/OER performance of CoNiP NA/NF-2 demonstrates that the optimization of surface/interface electronic structures can be realized in multicomponent metal phosphide alloys to improve the HER and/or OER performance.

## 2. Results and Discussion

### 2.1. Structural Characterization of Catalysts

As shown in Figure 1, the 3D CoNiP NA/NF nanoarrays were synthesized through a hydrothermal method followed by a phosphorization process. First, the CoNi-hydroxide nanoarrays were grown in situ on a pre-treated 3D microporous nickel foam substrate by the hydrothermal method. Second, CoNiP NA/NF was obtained by low-temperature phosphatizing reaction. The crystal structure of the as-prepared samples was characterized by XRD patterns. Appendix A presents the XRD patterns of CoNi-hydroxide NA/NF and CoNiP NA/NF with different Co/Ni ratios, respectively. It can be observed that all samples have three strong diffraction peaks, which are assigned to the metallic Ni of NF. The other peaks are attributed to CoNi-hydroxides (JCPDS: 00-040-0216), proving the successful preparation of basic hydroxide precursors [20,21]. The peak position of CoNi-hydroxide NA/NF (Co:Ni = 1:2) shifted to a lower angle compared with that of Ni-hydroxide NA/NF (Co:Ni = 0:3). This indicates the increase of the lattice constant, which is attributed to the incorporation of the object atom (Ni) into Ni-hydroxides. As the Co/Ni ratio increases, the trend of the peak shift increases, further demonstrating the above conclusions. This phenomenon proves the formation of CoNi-hydroxide alloys. For CoNiP NA/NF with different Co/Ni ratios, the peaks located at 40.8° and 45.3° are all well-matched with the characteristic peaks of the CoNiP alloy. Typically, the diffraction peaks located at 40.8° can be assigned to the (111) crystal plane of the CoNiP alloy [22,23].

SEM was firstly employed to illustrate the morphologies and microstructures of the as-synthesized CoNi-hydroxide NA/NF. Ni foam presents the smooth surface with three dimensional porous and crosslinked skeleton structures (Figure 2a). Co-hydroxide NA/NF nanoarrays are composed of nanowires (Figure 2b). Interestingly, with the Ni/Co ratio gradually increasing, the nanowire epitaxy grows and gradually transforms into nanosheets (Figure 2c–f). Typically, the CoNi-hydroxide NA/NF-2 sample presents a unique structure of the epitaxial growth of nanowires into nanosheets. The fine structure of the nanoarrays can be effectively adjusted by controlling the Co/Ni ratio, resulting in different surface properties. 

The morphology of the samples after phosphating was also observed by SEM. As shown in Figure 3a and Appendix A, the nanoarray structure of samples is almost maintained, proving its micro-structure stability. According to the TEM image, CoNiP NA/NF-2 is constituted of nanosheet epitaxial growth on nanowires (Figure 3b). HRTEM shows the typical fringe lattice of 0.220 nm, which can be indexed to the (111) plane of CoNiP (Figure 3c). EDS elemental mapping shows that Co, Ni, P, and O elements were uniformly dispersed on a morphology of nanosheets. Oxygen species may originate from the surface oxidation of the NiCoP phase. The above discussion further demonstrates the successful preparation of bimetallic phosphating alloys.

Appendix A presents the nitrogen adsorption–desorption isotherm and the corresponding pore size distribution of CoNiP NA/NF-2. The nitrogen adsorption–desorption isotherm presents a typical shape of the hysteresis loop, showing that abundant micropores and mesopores exist in CoNiP NA/NF-2. According to the calculated BET surface area, the CoNiP NA/NF-2 possesses a large surface area of 145.7 m^2^ g^−1^, corresponding to a wide mesopore size distribution in the range of 9~45 nm. The 3D cobalt-nickel phosphide nanoarray in situ grown on a Ni foam catalyst provided a larger surface area and porous structure, and thus a more favorable mass transfer process can be carried out, thereby manifesting enhanced catalytic performance.

XPS was employed to study the composition and surface chemical property of the prepared samples. The survey spectrum of CoNiP NA/NF with different Co/Ni ratios all present obvious Co 2p, Ni 2p and P 2p peaks (Figure 4a). Figure 4b shows the XPS fine spectra of Co 2p; the peaks of Co_2_P NA/NF appearing at 778.6 (2p_3/2_) and 793.5 (2p_1/2_) eV can be identified as Co^2+^, and the peaks appearing at 782.3 (2p_3/2_) and 798.7 (2p_1/2_) eV can be identified as Co^3+^. These peaks originate from the oxidized cobalt of Co-PO_x_ and Co-P bonding, respectively [24,25]. The high-resolution Ni 2p spectrum of Ni_2_P NA/NF exhibits characteristic peaks located at 853.4 and 869.7 eV, assigned to Ni^0^. The peaks at 857.0 and 874.6 eV can be attributed to Ni^δ+^ (2 < δ < 3), which are related to Ni-P and the oxidized nickel of Ni-PO_x_ (Figure 4c) [26]. It is worth noting that, compared to the Co 2p peaks of Co_2_P NA/NF and Ni 2p peaks of Ni_2_P NA/NF, the Co 2p and Ni 2p peaks of CoNiP NA/NF with different Co/Ni ratios all significantly moved to a lower binding energy, demonstrating that electrons of the Co-Ni binary metal tend to transfer to around P atoms in comparison with mono-metal, thereby facilitating adjustment of the surface electronic structure of bimetallic phosphating alloys. For P 2p spectra, the peaks at 129.5 and 130.4 eV correspond to P 2p_3/2_ and P 2p_1/2_ in metal phosphides. The peak at 134.1 eV is indexed to the oxidized phosphorus species originating from the surface oxidation and corresponding to the Co/Ni oxide state mentioned above. The peaks of CoNiP NA/NF with different Co/Ni ratios located at higher binding energy than that of Co_2_P NA/NF and Ni_2_P NA/NF indicate that more electrons are transferred to P atoms from bimetals (Figure 4d) [26,27,28]. XPS results indicate the strong electronic interaction between Co-Ni bimetals and P species.

### 2.2. Electrochemical Characterization for HER and OER

The electrocatalytic performance for HER of Co_2_P NA/NF, CoNiP NA/NF-1, CoNiP/NF-2, CoNiP/NF-3 and Ni_2_P NA/NF was investigated in 1.0 M KOH solution using a typical three-electrode system. Pt/C/NF and NF were also investigated as comparisons. Figure 5a displays the linear sweep voltammogram (LSV) curves of the as-prepared samples. Similar to the previous report, Pt/C/NF exhibits superior electrocatalytic activity for HER, with a small overpotential of 27 mV at a current density of 10 mA cm^−2^ (*η*_10_). For the bare NF, it shows undesired catalytic activity, with a large overpotential of 211 mV at *η*_10_, owing to its inert property (Figure 5b). In contrast, CoNiP NA/NF-2 presents remarkably improved HER activity, with a small overpotential of 162 mV at *η*_10_, smaller than Co_2_P NA/NF (177 mV) and Ni_2_P NA/NF (182 mV), demonstrating the increased activity of bimetallic phosphating alloys as compared to mono-metal phosphide. This precedes current reporting (Appendix A). CoNiP NA/NF-2 also shows lower overpotential than CoNiP NA/NF-1 (167 mV) and CoNiP NA/NF-3 (173 mV), which can be attributed to its optimal surface electronic structure being regulated by Ni/Co ratio. The Tafel slope is employed to evaluate the HER kinetics of the samples. As shown in Figure 5c, the Tafel slope of the CoNiP NA/NF-2 is fitted as 114.3 mV dec^−^^1^, which is obviously smaller than that of Co_2_P/NF (153.1 mV dec^−1^), CoNiP NA/NF-1 (127.4 mV dec^−1^), CoNiP NA/NF (124.5 mV dec^−1^), Ni_2_P/NF (155.6 mV dec^−1^) and NF (263.5 mV dec^−1^), demonstrating the fast HER kinetics of CoNiP NA/NF-2. Such favorable kinetics of CoNiP NA/NF-2 are related to the unique nanoarray structure and synergy of bimetallic components. The structural advantage of CoNiP NA/NF-2 can be further proven by the highest double-layer capacitance (C_dl_) of CoNiP/NF (27.8 mF cm^−2^), which is superior to that of Co_2_P NA/NF (7.2 mF cm^−2^), CoNiP NA/NF-1 (7.2 mF cm^−2^), CoNiP NA/NF-3 (9.7 mF cm^−2^), Ni_2_P NA/NF (14.9 mF cm^−2^) and NF (1.5 mV dec^−1^). Therefore, CoNiP NA/NF-2 possesses the optimal mass/charge transfer process and more active sites (Figure 5d). Electrochemical impedance spectroscopy (EIS) was employed to further explore the charge-transfer mechanism (Figure 5e). The CoNiP NA/NF-2 exhibits the smallest R_ct_ values of 4.97 Ω, which are smaller than Co_2_P/NF (7.42 Ω), CoNiP NA/NF-1 (7.27 Ω), CoNiP/NF-3 (7.77 Ω), and Ni_2_P/NF (12.79 Ω), implying its faster charge transfer process. The long-term HER electrochemical durability was studied by a multi-current step chronopotentiometric curve (Figure 5f). The current density from 10 to 190 mA cm^−2^ and from 190 to 10 mA cm^−2^ observed in each segment was barely changed, demonstrating excellent stability of the CoNiP NA/NF-2 sample.

The electrocatalytic performance of the as-prepared catalysts for OER were investigated. The CoNiP NA/NF-2 exhibits excellent electrocatalytic activity for OER, with the overpotentials of 499 mV at a current density of 100 mA cm^−2^ (*η*_100_), which was close to the overpotentials of commercial RuO_2_ (474 mV) and much superior to those of Co_2_P NA/NF (518 mV), CoNiP NA/NF-1 (547 mV), CoNiP NA/NF-3 (539 mV), Ni_2_P NA/NF (598 mV) and NF (604 mV) (Figure 6a,b). The CoNiP NA/NF-2 also possesses a superior Tafel slope of 79.5 mV dec^−1^ and the highest C_dl_ of 10.8 mF cm^−2^. The above results demonstrate a more rapid water oxidation of CoNiP NA/NF-2 and more exposed active sites. The CoNiP NA/NF-2 also exhibits the smallest semicircle, with R_ct_ values of 4.97 Ω, demonstrating its superior charge transmission and lower charge-transfer resistance (Figure 6e). The long-term electrochemical durability for OER was also studied using a multi-current step chronopotentiometric curve. A steady current density of the CoNiP NA/NF-2 can be observed (Figure 6f).

Inspired by the excellent catalytic performance of CoNiP NA/NF-2 for HER and OER, the overall water splitting performance was also tested in a two-electrode system. As shown in Figure 7a, when CoNiP NA/NF-2 serves as both cathode and anode, it needs only 1.613 V to reach the current density of 10 mA cm^−2^. This is comparable to the reported TMP-based bifunctional electrocatalysts (Appendix A). Meanwhile, the CoNiP NA/NF-2||CoNiP NA/NF-2 system also presents outstanding durability, with a slight amplitude of potential variation for 24 h at 10 mA cm^−2^ (Figure 7b). The faradaic efficiency for hydrogen production up to 99.6% over the CoNiP NA/NF-2||CoNiP NA/NF-2 system for 1 h during the overall water splitting is shown in Figure 7c,d. In addition, the volume ratio of hydrogen to oxygen is close to 2:1, which is consistent with the theoretical value.

### 2.3. Theoretical Calculation

To further analyze the excellent catalytic activity of CoNiP NA/NF for HER and OER, a Density Functional Theory (DFT) calculation was conducted. The surface (001) of CoNiP, Co_2_P, and Ni_2_P were constructed as shown in Appendix A. The free energy of H-atom (ΔG_H*_) absorbed at the active sites of CoNiP, Co_2_P and Ni_2_P is shown in Figure 8a. It can be seen that the surface of CoNiP has the lowest ΔG_H*_ (−0.37 eV), which is superior to that of pure Co_2_P (−0.64 eV) and Ni_2_P (−0.42 eV), indicating that the introduction of Co species into Ni_2_P can greatly promote the intrinsic HER catalytic activity, and the surface of CoNiP (001) was more conducive to promoting the HER [29]. The electronic structure after adsorption of H* on the catalyst surface was calculated. The electronic structure was also investigated. As shown in Figure 8b and Appendix A, the charge density of CoNiP shows the overt charge accumulation around H* and charge depletion at the surface of CoNiP. Correspondingly, the weaker charge transfer on Ni_2_P and Co_2_P than CoNiP demonstrates that the bimetallic component is beneficial to regulate the electronic energy state of Ni_2_P. The results of the electrostatic potential distribution after adsorption of H*, cross section of charge density, and the corresponding electron localization function also confirm this (Figure 9a–f and Appendix A).

## 3. Materials and Methods

### 3.1. Preparation of Cobalt/Nickel-Hydroxide Nanoarray Precursor Grown on Nickel Foam (CoNi-Hydroxides NA/NF)

First, the nickel foam (NF, 2 × 3 cm) was treated with 1.0 M HCl solution, ethanol and distilled water for 20 min by ultrasonication to remove residual surface oxide. For the preparation of CoNi-hydroxide NA/NF-2 precursor with a Co/Ni molar ratio of 1:1, Co(NO_3_)_2_·6H_2_O (1.5 mmol), Ni(NO_3_)_2_·6H_2_O (1.5 mmol), NH_4_F (8 mmol) and urea (15 mmol) were first dissolved in 30 mL deionized water. After continuously stirring for 30 min, the mixed solution was transferred into a Teflon-lined stainless autoclave (50 mL) (Anhui Kemi Machinery Technology Co., Ltd. Hefei, China), and a pre-treated NF was immersed into the autoclave and heated at 120 °C for 6 h. After cooling down to room temperature, the hydroxide precursor was thoroughly rinsed with water (three times) and ethanol (one time) and then dried at 60 °C for 6 h. Similarly, the other CoNi-hydroxide NA/NF precursor with a Co/Ni molar ratio of 2:1 and 1:2 were also prepared by a similar procedure (the total dosage of Co(NO_3_)_2_·6H_2_O and Ni(NO_3_)_2_·6H_2_O was 3 mmol) and named as CoNi-hydroxide NA/NF-1 and CoNi-hydroxide NA/NF-3. Co-hydroxide NA/NF and Ni-hydroxide NA/NF precursors were also prepared with a similar approach, except that only 3 mmol of Co(NO_3_)_2_·6H_2_O or Ni(NO_3_)_2_·6H_2_O was added.

### 3.2. Preparation of Cobalt/Nickel Nanoarrays Grown on Nickel Foam (CoNiP NA/NF-2)

For the synthesis of CoNiP NA/NF, the CoNi-hydroxide NA/NF and 300 mg of sodium hypophosphite were applied on both sides of porcelain vessels in a tube furnace. After flushing with nitrogen under a flow rate of 700 sccm for about 30 min, the tube furnace was heated at 300 °C for 2 h, with a heating rate of 2 °C min^−1^ under with a flow rate of 40 sccm. The prepared samples were named CoNiP NA/NF-1 (Co/Ni = 2:1), CoNiP NA/NF-2 (Co/Ni = 1:1), CoNiP NA/NF-3 (Co/Ni = 1:2), Co_2_P NA/NF (Co/Ni = 3:0) and Ni_2_P NA/NF (Co/Ni = 0:3).

### 3.3. Characterization

The microstructure of the materials was characterized by field emission scanning electron microscope (FESEM, JEOL JSM7500 F) and transmission on electron microscope (TEM, FEI S-TWIN) at 200 kV. High angle annular dark-field scanning transmission electron microscopy (HAADF-STEM) and energy dispersive X-ray spectroscopy (EDS) maps were performed on a FEI Talos, F200S instrument. The powder X-ray diffraction (XRD) tests were performed on a PANalytcal X’Pert PRO instrument (Cu-K*α* radiation, 40 kV, 40 mA, λ = 0.154 nm). X-ray photoelectron spectroscopy (XPS) was carried out on a PHI quantera SXM spectrometer using Al-K*α* radiation.

### 3.4. Electrochemical Measurement

The electrochemical measurements were measured on a CHI660E electrochemical working station using a conventional three-electrode system at room temperature. The as-prepared CoNiP NA/NF serves as the working electrode, and the graphite rod and mercuric oxide electrode (Hg/HgO) work as the counter electrode and reference electrode, respectively. The catalytic activity towards HER and OER was examined in a 1.0 M KOH electrolyte solution. For over-water splitting, the catalyst was measured via a two-electrode cell configuration, and the as-synthesized CoNiP NW/NF served as a cathode and an anode.

The HER and OER performance of the electrocatalysts was measured by linear sweep voltammetry (LSV) with a scan rate of 5 mV s^−1^. All of the potentials were calibrated to the reversible hydrogen electrode (RHE) according to the following equation:E (RHE) = E (Hg/HgO) + 0.098 + 0.059pH 

Electrode surface area:1 cm × 1 cm = 1 cm^2^

Electrochemical impedance spectroscopy (EIS) was tested at the voltage corresponding to current density of 10 mA cm^−2^ from 0.1 to 10^5^ Hz. Cyclic voltammetry (CV) measurement scanning was used to calculate the double layer capacitance (C_dl_) at non-faradaic potential regions with different scan rates, including 15, 20, 25, 30, 35 and 40 mV s^−1^. Then, the curve was plotted with regards to the current density and the scan rate at a certain potential, and the slope of the curves was the value of C_dl_. The electrochemical active surface area (ECSA) of samples was proportional to the value of C_dl_.

The gas generated was collected by drainage gas collection. All of the measurements were corrected without iR compensation.

In case of OER, the entire progress can be generally summarized as four elementary reaction models (the OER process is considered via a four-proton-coupled electron transfer mechanism), which consist *OH, *O, *OOH and O_2_ [30,31], as presented in Figure 8c,d. To further evaluate the interaction between the intermediates and catalyst surface, the adsorption energies of the intermediates (ΔG_*OH_, ΔG_*O_, ΔG_*OOH_ and ΔG_O2_) were determined and are shown in Figure 8e. The ΔG_*OOH_ values of CoNiP and Ni_2_P were 2.81 and 3.02 eV, respectively, demonstrating the formation of *OOH as the rate-limiting step in whole OER process. The small energy required to generate *OOH suggests that the introduction of Co species into Ni_2_P is conducive to alkaline OER activity. As shown in Figure 8f and Appendix A, the charge density of CoNiP and Ni_2_P shows the overt charge accumulation around H* and charge depletion at the surface of CoNiP and Ni_2_P. The results of the electrostatic potential distribution after the adsorption of H*, cross section of charge density difference, and the corresponding electron localization function also confirm this (Figure 9g–l). In addition, we calculated the density of the states of Ni and P in CoNiP and Ni_2_P (Appendix A). An obvious right shift of TDOS is observed on CoNiP compared to bare Ni_2_P, which results in a stronger binding of the intermediates.

## 4. Conclusions

In summary, a series of novel 3D cobalt-nickel phosphide nanoarrays in situ grown on Ni foam catalyst were developed with tunable surface properties and unique micro-morphology. The optimal CoNiP NA/NF-2 afforded the current density of 10 mA cm^−2^ at a low overpotential of 162 for HER and 499 mV (100 mA cm^−2^) for OER, corresponding with low Tafel slopes of 114.3 and 79.5 mV dec^−1^, respectively, as well as robust durability. It needed only a cell potential of 1.613 V to achieve the current density of 10 mA cm^−2^. The nanosheet array structure offers more available active sites and high charge-transfer kinetics for the HER/OER process. DFT calculations demonstrate that the optimal catalyst surface modulated by the Ni/Co ratio facilitates the adsorption of absorbed H* and *OOH intermediates, resulting in an optimized energy barrier for HER and OER. 

## Figures and Tables

**Figure 1 ijms-23-05290-f001:**
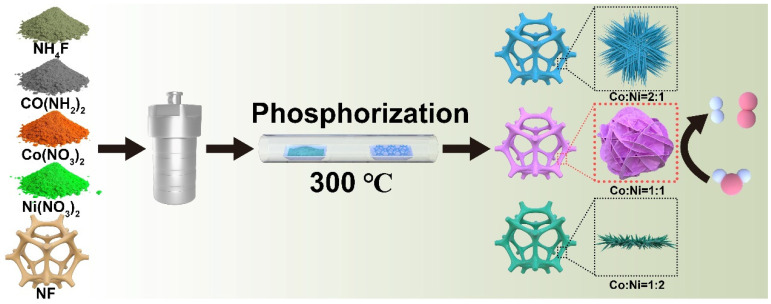
Schematic diagram of the fabrication of CoNiP/NA/NF nanowires.

**Figure 2 ijms-23-05290-f002:**
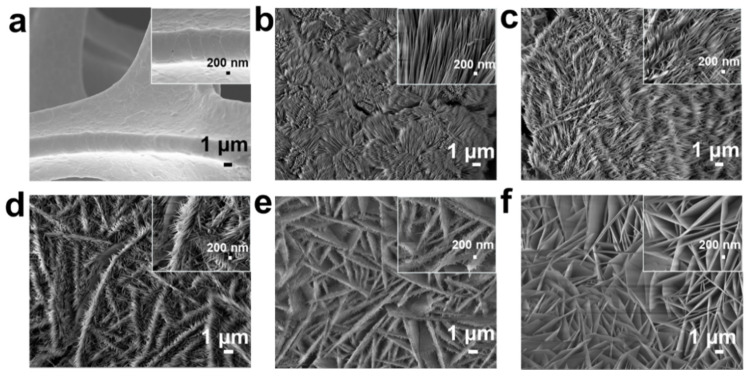
SEM images of (**a**) pre-treated Ni foam, (**b**) Co-hydroxide NA/NF, (**c**) CoNi-hydroxide NA/NF-1, (**d**) CoNi-hydroxide NA/NF-2, (**e**) CoNi-hydroxide NA/NF-3 and (**f**) Ni-hydroxide NA/NF.

**Figure 3 ijms-23-05290-f003:**
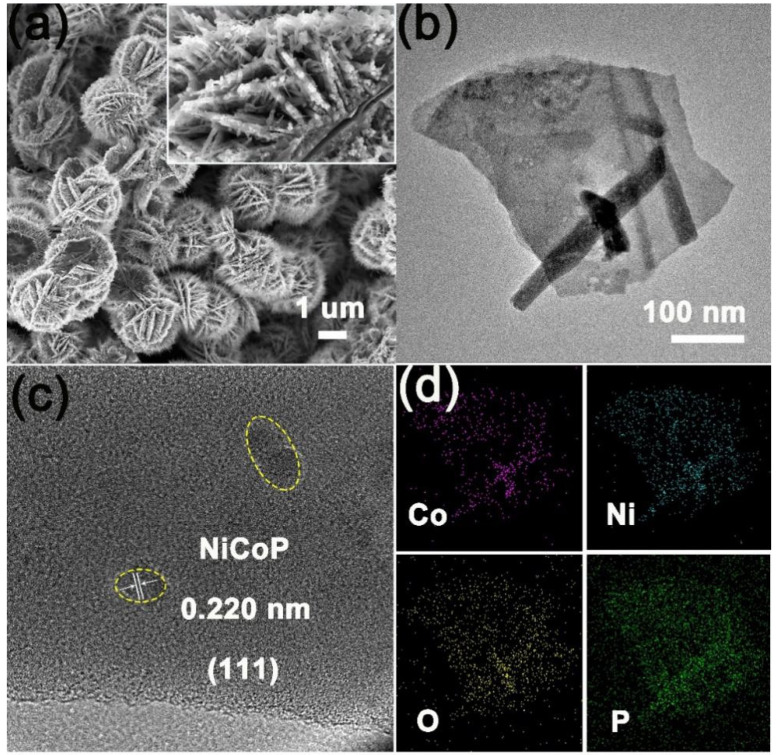
(**a**) SEM image of CoNiP NA/NF-2. (**b**) TEM image of CoNiP obtained from CoNiP NA/NF-2. Corresponding (**c**) HRTEM image and (**d**) EDS elemental mapping of Co, Ni, O and P.

**Figure 4 ijms-23-05290-f004:**
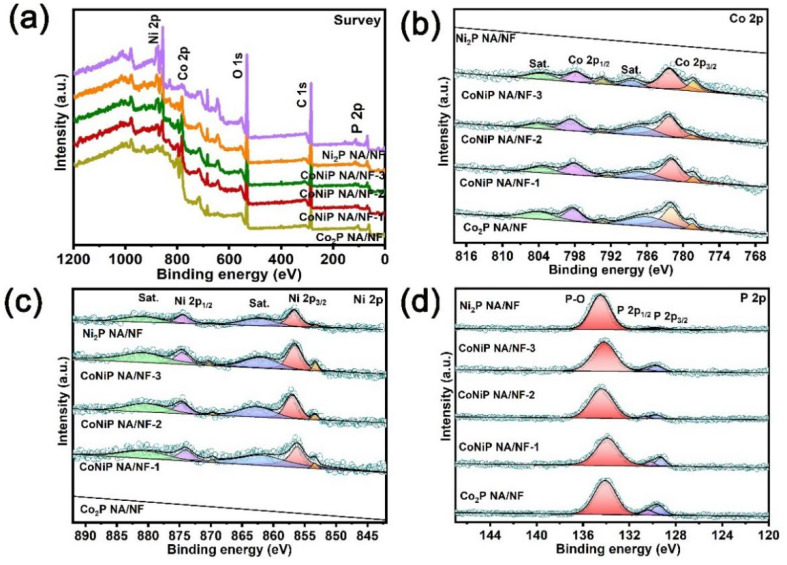
(**a**) XPS survey spectra of CoNiP NA/NF-1, CoNiP NA/NF-2, CoNiP NA/NF-3, Co_2_P NA/NF and Ni_2_P NA/NF. (**b**) High-resolution XPS spectra of (**b**) Co 2p (**c**) Ni 2p and (**d**) P 2p of samples.

**Figure 5 ijms-23-05290-f005:**
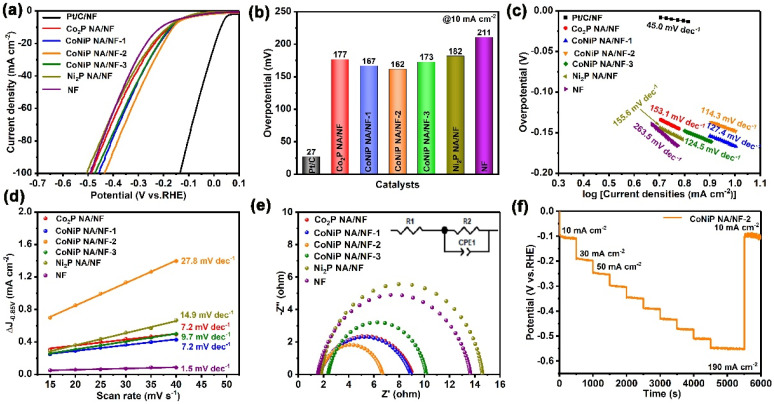
The HER catalytic performances of samples in 1.0 M KOH. (**a**) HER polarization curves over the various samples at a scan rate of 5 mV s^−1^. (**b**) The corresponding overpotentials at the current density of 10 mA cm^−2^. (**c**) The corresponding Tafel slopes. (**d**) The plots of the current density versus the scan rate for the catalysts. (**e**) The Nyquist curves of the obtained catalysts (the inset is the equivalent circuit). (**f**) Multi-current step chronopotentiometric curves.

**Figure 6 ijms-23-05290-f006:**
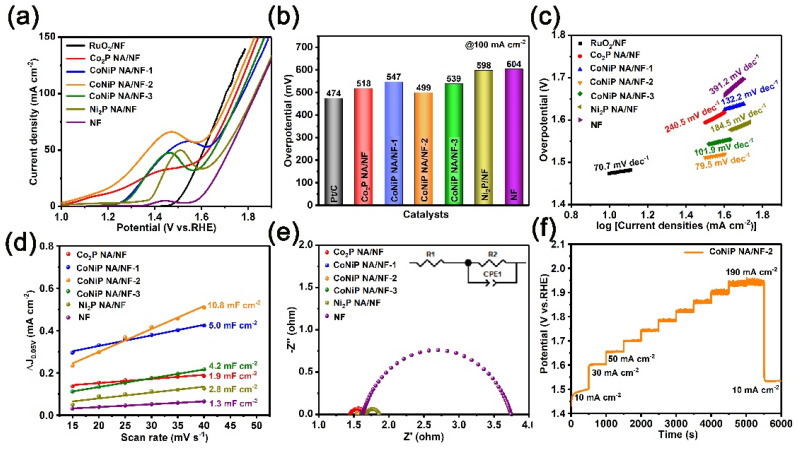
The OER catalytic activity of samples in 1.0 M KOH. (**a**) LSV polarization curves, (**b**) the corresponding overpotentials at *η*_10_, (**c**) Tafel slopes, (**d**) the C_dl_ and (**e**) the Nyquist curves (the inset is the equivalent circuit) of as-prepared samples. (**f**) Multi-current step chronopotentiometric curves of CoNiP NA/NF.

**Figure 7 ijms-23-05290-f007:**
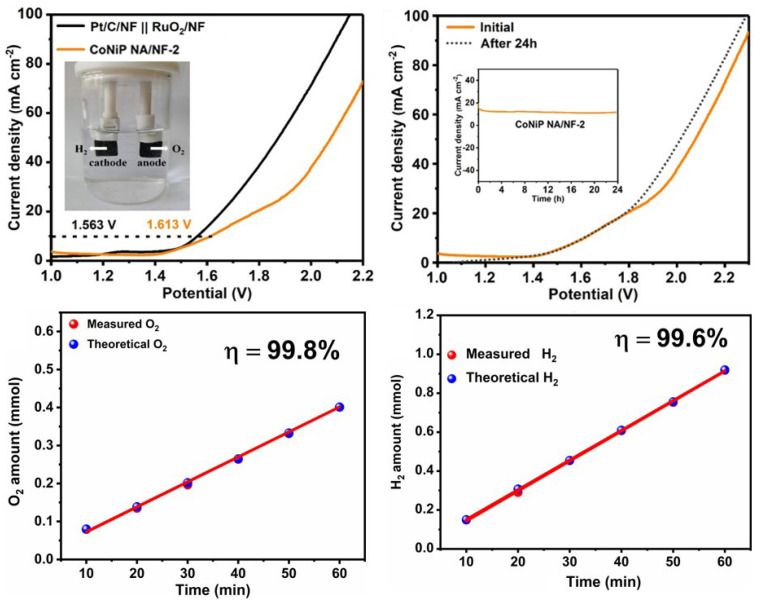
(**a**) LSV polarization curves of CoNiP NA/NF-2 for overall water splitting. (**b**) Chronopotentiometry test of CoNiP NA/NF-2 at a current density of 10 mA cm^−2^ in a two-electrode electrolyzer for 24 h. The faradaic efficiency of (**c**) O_2_ generation over CoNiP NA/NF-2 electrode at 10 mA cm^−2^ and (**d**) H_2_ generation over CoNiP NA/NF-2 electrode at 10 mA cm^−2^ for 1 h.

**Figure 8 ijms-23-05290-f008:**
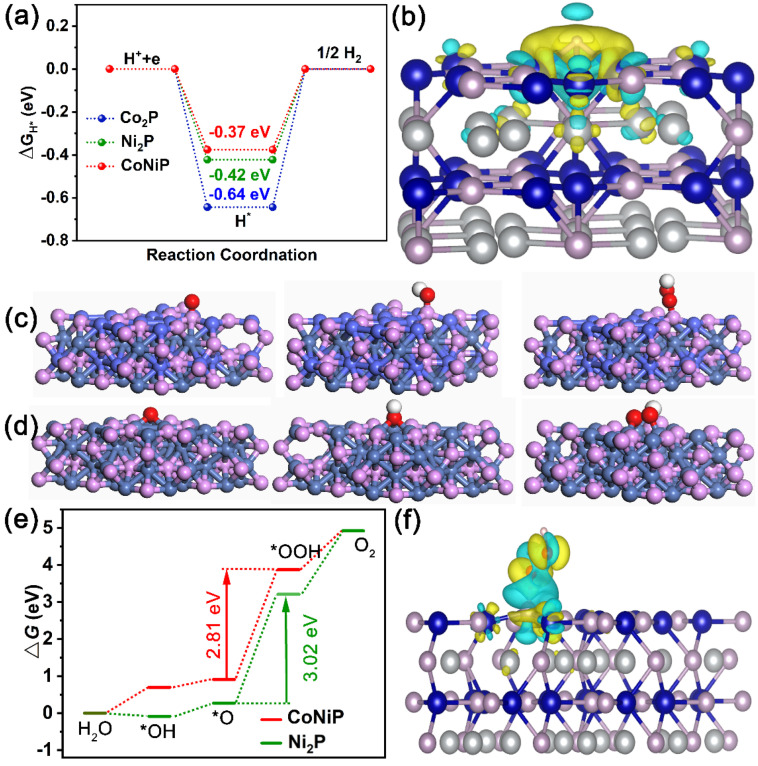
(**a**) The ΔG_H*_ values for CoNiP, Co_2_P and Ni_2_P. (**b**) Charge density difference plots at the surface of CoNiP for HER. Structures of *OH, *O and *OOH intermediates of the OER process on (**c**) CoNiP and (**d**) Ni_2_P. (**e**) Gibbs free energy diagrams for OER processes. (**f**) The charge-density difference of plots at the surface of CoNiP for OER.

**Figure 9 ijms-23-05290-f009:**
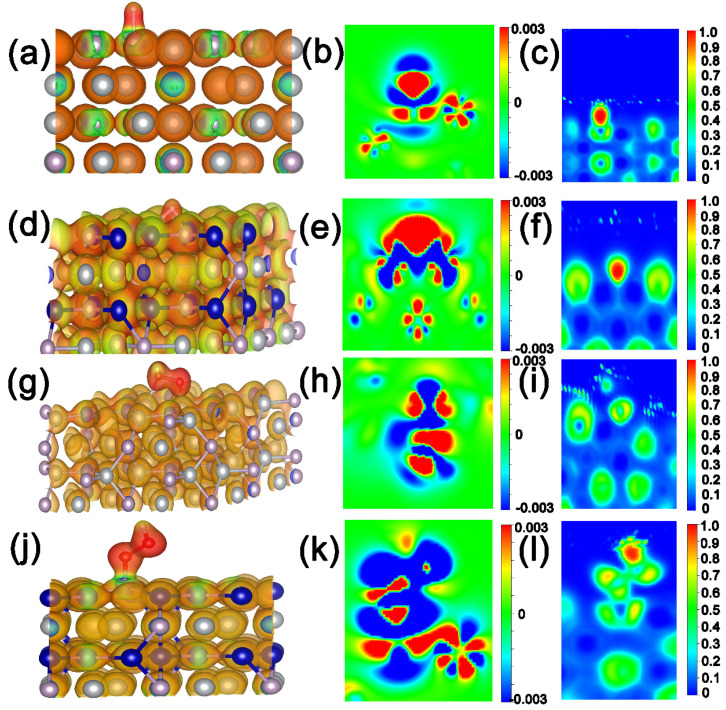
Electrostatic potential distribution after adsorption of H* on the (**a**) Ni_2_P and (**d**) CoNiP surface. The corresponding cross section of charge density difference of (**b**) Ni_2_P, (**e**) CoNiP and (**d**) Ni_2_P. The corresponding electron localization function (ELF) of (**c**) Ni_2_P and (**f**) CoNiP. Electrostatic potential distribution after adsorption of *OOH on the (**g**) Ni_2_P and (**j**) CoNiP surface. The corresponding cross section of charge density difference of (**h**) Ni_2_P, (**k**) CoNiP. The corresponding electron localization function (ELF) of (**i**) Ni_2_P and (**l**) CoNiP.

## Data Availability

The data presented in this study are available on request from the corresponding author.

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
