# Peer review of "Surface Modulation of 3D Porous CoNiP Nanoarrays In Situ Grown on Nickel Foams for Robust Overall Water Splitting"

_ijms, 2022, doi:10.3390/ijms23105290_

Round 1

Reviewer 1 Report

This manuscript deals with the electrocatalytic activity for the hydrogen evolution (HER) and oxygen evolution (OER) reactions of porous CoNiP layers supported by Ni frameworks. The authors have decorated Ni frameworks with binary cobalt-nickel phosphides and systematically studied the effect of the relative content of Ni/Co. The data that the authors present supports their conclusions and the results are good, however there are some very notable measurements missing and some discrepancies in the results in the manuscript for it to be acceptable for publication.

  • What is the purpose in the synthesis of urea and NH4F? Do N and F stay in the catalyst material (a small peak can be observed in the XPS spectra for the nitrogen area) and how do they affect the activity of the catalysts? More light could be shed on this issue by bulk elemental analysis such as XRF or ICP-MS.
  • Interestingly, even though the first word in the title of the article is “porous”, no porosity analysis has been conducted by the authors. This critical measurement should definitely be included into the manuscript.
  • The quality of Figures is low. This is likely due to compression when working in Microsoft Word and can be turned off from Options -> Advanced -> Image Size and Quality.
  • The CoNiP has much larger overpotentials for both the HER and the OER than the state-of-the-art materials the authors have presented. However, in the two-electro electrolyzer, the CoNiP performs better. This might signify an issue with the electrode preparation for this measurement and should be repeated.
  • The LSV curves for water splitting are done with the CoNiP NA/NF-2 material, while the stability testing is done with the CoNiP NA/NF catalyst, and the efficiency measurements are done with a RuCoP catalyst, which is not mentioned elsewhere, nor is it described how this material was synthesized. Please use the same CoNiP NA/NF-2 for all the tests.
  • Some topical articles on other alternatives to Pt and RuO2 should be added to references,: R. Li, J. Zang, W. Li, J. Li, Q. Zou, S. Zhou, J. Su, Y. Wang, ChemSusChem. 13 (2020) 3718–3725; X. Du, J. Fu, X. Zhang, NiCo2O4@NiMoO4 Supported on Nickel Foam for Electrocatalytic Water Splitting, ChemCatChem. 10 (2018) 5533–5540; A. Remmel, S. Ratso, G. Divitini, M. Danilson, V. Mikli, M. Uibu, J. Aruväli, I. Kruusenberg, ACS Sustain. Chem. Eng. 10 (2021) 134–145; T. Wang, Q. Zhou, X. Wang, J. Zheng, X. Li, J. Mater. Chem. A. 3 (2015) 16435–16439; T. Wang, X. Wang, Y. Liu, J. Zheng, X. Li, Nano Energy. 22 (2016) 111–119; S. Ratso, P.R. Walke, V. Mikli, J. Ločs, K. Šmits, V. Vītola, A. Šutka, I. Kruusenberg, Green Chem. 23 (2021) 4435–4445; Ullah, W. Zhao, X. Lu, C.J. Oluigbo, S.A. Shah, M. Zhang, J. Xie, Y. Xu, Electrochim. Acta. 298 (2019) 163–171.

Author Response

Dear respected editor and reviewers,

Thank you for your letter to inform us your welcome decision and the reviewer’s positive comments on our manuscript (ID: ijms-1680621). According to the reviewer’s comments and your suggestions, we have carefully revised our manuscript. Now we are resubmitting the revised version with a detailed point-to-point response to the reviewer’s comments, and list of revisions made. We hope that the revised manuscript is suitable for publication in your journal- International Journal of Molecular Sciences. If there are any questions about the revision, please feel free to contact me at any time. We thank the reviewers for their correct comments and look forward to hearing good news from you regarding our revised manuscript.

Best wishes!

Sincerely yours,

Prof. Zhikun Peng

Reviewer: 1

Comments:

This manuscript deals with the electrocatalytic activity for the hydrogen evolution (HER) and oxygen evolution (OER) reactions of porous CoNiP layers supported by Ni frameworks. The authors have decorated Ni frameworks with binary cobalt-nickel phosphides and systematically studied the effect of the relative content of Ni/Co. The data that the authors present supports their conclusions and the results are good, however there are some very notable measurements missing and some discrepancies in the results in the manuscript for it to be acceptable for publication.

Question 1. What is the purpose in the synthesis of urea and NH4F? Do N and F stay in the catalyst material (a small peak can be observed in the XPS spectra for the nitrogen area) and how do they affect the activity of the catalysts? More light could be shed on this issue by bulk elemental analysis such as XRF or ICP-MS.

Response: Thanks for your constructive question. Urea and ammonium fluoride were added to form the metal coordination compound deposited on foam nickel. A solution of ammonium fluoride has slightly acidic properties, and urea makes this solution less acidic. It apparently promotes the deposition of complexes. Unfortunately, it is difficult accurately quantify the F and N elements. The implementation of many characterization methods is limited due to the epidemic situation. The content of nitrogen and fluorine is roughly estimated by the XPS survey spectra of samples. Very weak peak signal is observed for N 1s (at about 398.6 eV). Therefore, the effect of trace amout of N on catalytic performance is negligible. The XPS survey spectra of samples show the peak of F 1s located at 684.25 eV. As for F species, Ji and co-works have developed a new type of reconfigurable fluoride (such as CoF2) pre-catalysts, with ultra-fast and in-depth self-reconstruction, substantially promoting HER activity. The alkaline electrolyte triggers rapid F leaching and supplies an immediate complement of OH−  to form amorphous �-Co(OH)2 which rapidly transforms into �-Co(OH)2. The bias voltage promotes amorphous crystallization and accelerates the reconstruction process. These endow the generation of mono-component and crystalline �-Co(OH)2 with a loose and defective structure, leading to an ultra-low overpotential and super long-term stability. Thus, the presence of fluorine in CoNi-NA/NF may contribute to the electrocatalytic overall water splitting.

Fig. 4 (a) XPS survey spectra of CoNiP NA/NF-1, CoNiP NA/NF-2, CoNiP NA/NF-3, Co2P NA/NF, and Ni2P NA/NF, respectively.

[R1] Ji, P.; Yu, R.; Wang, P.; Pan, X.; Jin, H.; Zheng, D.; Mu, S, Ultra-Fast and In-Depth Reconstruction of Transition Metal Fluorides in Electrocatalytic Hydrogen Evolution Processes. Adv. Sci. 2022, 9, 2103567.

Question 2. Interestingly, even though the first word in the title of the article is “porous”, no porosity analysis has been conducted by the authors. This critical measurement should definitely be included into the manuscript.

Response: Thanks for your constructive question. Figure S3a-b presents the nitrogen adsorption-desorption isotherm and the corresponding pore size distribution of CoNiP NA/NF-2. The nitrogen adsorption-desorption isotherm presents typical shape of the hysteresis loop, manifesting that abundant micropores and mesopores are existed in CoNiP NA/NF-2. According to the calculated BET surface area, the CoNiP NA/NF-2 possesses large surface area of 145.7 m2 g-1, corresponding with a wide mesopore size distribution in the range of 9~45 nm. The 3D cobalt-nickel phosphide nanoarray in situ grown on Ni foam catalyst has provided larger surface area and porous structure, thus a more favorable mass transfer processes can be carried out, thereby manifesting enhanced catalytic performance.

Figure S3 (a) Nitrogen adsorption-desorption isotherms and (b) corresponding pore size distribution  of CoNiP NA/NF-2

Question 3. The quality of Figures is low. This is likely due to compression when working in Microsoft Word and can be turned off from Options -> Advanced -> Image Size and Quality.

Response: Thank you for your advice and patient guidance. The quality of figures has been improved according to your  guidance.

Question 4. The CoNiP has much larger overpotentials for both the HER and the OER than the state-of-the-art materials the authors have presented. However, in the two-electro electrolyzer, the CoNiP performs better. This might signify an issue with the electrode preparation for this measurement and should be repeated.

Response: Thank you for your careful review. The names of CoNiP NA/NF-2  and Pt/C/NF || RuO2 in Figure 7a are wrongly swapped due to our imprudence.  It has been revised in detail in the manuscript, and we have checked the entire manuscript to avoid such errors before resubmitting.

Figure 7 (a) LSV polarization curves of CoNiP NA/NF-2 for overall water splitting. (b Chronopotentiometry test of CoNiP NA/NF-2 at a current density of 10 mA cm-2 in a two-electrode electrolyzer for 24 h. The faradaic efficiency of (c) O2 generation over CoNiP NA/NF-2 electrode at 10 mA cm-2 and (d) H2 generation over CoNiP NA/NF-2 electrode at 10 mA cm-2 for 1 h, respectively.

Question 5. The LSV curves for water splitting are done with the CoNiP NA/NF-2 material, while the stability testing is done with the CoNiP NA/NF catalyst, and the efficiency measurements are done with a RuCoP catalyst, which is not mentioned elsewhere, nor is it described how this material was synthesized. Please use the same CoNiP NA/NF-2 for all the tests.

Response: Thank you for your careful review and useful suggestions. Due to our imprudence, it led to the severe error of inconsistent names of our catalyst. In fact, on the basis of the previous tests, the LSV test for water splitting, stability test, and  efficiency measurements all revolve around CoNiP NA/NF-2 sample. This section has been revised in detail in the manuscript, and we have checked the entire manuscript to avoid such errors before resubmitting.

Figure 7 (a) LSV polarization curves of CoNiP NA/NF-2 for overall water splitting. (b) Chronopotentiometry test of CoNiP NA/NF-2 at a current density of 10 mA cm-2 in a two-electrode electrolyzer for 24 h. The faradaic efficiency of (c) O2 generation over CoNiP NA/NF-2 electrode at 10 mA cm-2 and (d) H2 generation over CoNiP NA/NF-2 electrode at 10 mA cm-2 for 1 h, respectively.

Inspired by the excellent catalytic performance of CoNiP NA/NF-2 for HER and OER, the overall water splitting performance was also tested in a two-electrode system. As shown in Figure 7a, When CoNiP NA/NF-2 serves as both cathode and anode, it needs only 1.613 V to reach the current density of 10 mA cm-2. It is comparable to the reported TMPs-based bifunctional electrocatalysts (Table S2). Meanwhile, CoNiP NA/NF-2||CoNiP NA/NF-2 system also presents outstanding durability with a slight amplitude of potential variation for 24 h at 10 mA cm-2 (Figure 7b). The faradaic efficiency for hydrogen production up to 99.6% over CoNiP NA/NF-2||CoNiP NA/NF-2 system for 1 h during overall water splitting Figure 7c-d. In addition, the volume ratio of hydrogen to oxygen is close to 2:1, consistent with the theoretical value.

Question 6. Some topical articles on other alternatives to Pt and RuO2 should be added to references,: R. Li, J. Zang, W. Li, J. Li, Q. Zou, S. Zhou, J. Su, Y. Wang, ChemSusChem. 13 (2020) 3718–3725; X. Du, J. Fu, X. Zhang, NiCo2O4@NiMoO4 Supported on Nickel Foam for Electrocatalytic Water Splitting, ChemCatChem. 10 (2018) 5533–5540; A. Remmel, S. Ratso, G. Divitini, M. Danilson, V. Mikli, M. Uibu, J. Aruväli, I. Kruusenberg, ACS Sustain. Chem. Eng. 10 (2021) 134–145; T. Wang, Q. Zhou, X. Wang, J. Zheng, X. Li, J. Mater. Chem. A. 3 (2015) 16435–16439; T. Wang, X. Wang, Y. Liu, J. Zheng, X. Li, Nano Energy. 22 (2016) 111–119; S. Ratso, P.R. Walke, V. Mikli, J. Ločs, K. Šmits, V. Vītola, A. Šutka, I. Kruusenberg, Green Chem. 23 (2021) 4435–4445; Ullah, W. Zhao, X. Lu, C.J. Oluigbo, S.A. Shah, M. Zhang, J. Xie, Y. Xu, Electrochim. Acta. 298 (2019) 163–171.

Response: Thank you for your constructive suggestions. These literatures are very helpful in improving the quality of our manuscript and have been cited as Ref. 1, 7, 8, 9, 11, 12, 14  

[1] Ratso, S.; Walke, P. R.; Mikli, V.; Ločs, J.; Šmits, K.; Vītola, V.; Kruusenberg, I.; CO2 turned into a nitrogen doped carbon catalyst for fuel cells and metal–air battery applications. Green Chem. 2021, 23, 4435-4445.

[7] Wang, T.; Zhou, Q.; Wang, X.; Zheng, J.; Li, X., MOF-derived surface modified Ni nanoparticles as an efficient catalyst for the hydrogen evolution reaction. J. Mater. Chem. A 2015, 32, 16435-16439.

[8] Wang, T.; Wang, X.; Liu, Y.; Zheng, J.; Li, X., A highly efficient and stable biphasic nanocrystalline Ni–Mo–N catalyst for hydrogen evolution in both acidic and alkaline electrolytes, Nano Energ. 2016, 22, 111-119.

[9] Li, R.; Zang, J.; Li, W.; Li, J.; Zou, Q.; Zhou, S.; Wang, Y., Three‐Dimensional Transition Metal Phosphide Heteronanorods for Efficient Overall Water Splitting. ChemSusChem2020, 13, 3718-3725.

[11] Ullah, N.; Zhao, W.; Lu, X.; Oluigbo, C. J.; Shah, S. A.; Zhang, M.; Xu, Y. In situ growth of M-MO (M= Ni, Co) in 3D graphene as a competent bifunctional electrocatalyst for OER and HER. Electrochim. Acta 2019, 298, 163-171.

[12] Du, X.; Fu, J.; Zhang, X., NiCo2O4@ NiMoO4 supported on nickel foam for electrocatalytic water splitting. ChemCatChem, 201810, 5533-5540.

[14] Remmel, A. L.; Ratso, S.; Divitini, G.; Danilson, M.; Mikli, V.; Uibu, M.; Kruusenberg, I., Nickel and Nitrogen-Doped Bifunctional ORR and HER Electrocatalysts Derived from CO2ACS Sus. Chem. Eng. 2021.

Reviewer: 2

Comments:

In the present manuscript, the authors assess the HER and OER performance of CoNiP nanoarrays grown on Ni foam. The Co, Ni or NiCo nanoarrays were prepared on Ni foam using hydrothermal method and then the samples were treated in P to form CoNiP nanoarrays. The authors have reported improved activity in HER and OER regions of linear sweep voltammetry. The authors have used theoretical and experimental approaches to investigate the systems and the work is interesting to the readers. However, the novelty of the work is not well-defined, crucial experimental data and discussions (listed below) are missing in the manuscript. Therefore, the current version of the manuscript is not suitable for publication. 

Question 1. The motivation and novelty of the work have not been clearly defined in the abstract. The authors should emphasize these points in the abstract. (abstract,introduction)

Response: Thank you for your constructive suggestions. The motivations and innovations of the manuscript  have been redistilled and highlighted in the abstract and introduction as follows:Abstract: Careful designing of nanostructures and multi-compositions of non-noble metal based electrocatalysts for highly efficient electrocatalytic hydrogen and oxygen evolution reaction (HER and OER) is of great significance to realize sustainable hydrogen release. Herein, bifunctional electrocatalysts of the three-dimensional (3D) cobalt-nickel phosphide nanoarray in situ grown on nickel foams (CoNiP NA/NF) were synthesized through a facile hydrothermal method followed by phosphorization. Due to the unique self-template nanoarray structure and tunable multicomponent system, the CoNiP NA/NF samples presents exceptional activity and durability for HER and OER. The optimized sample of CoNiP NA/NF-2 afforded a current density of 10 mA cm-2 at a low overpotential of 162 mV for HER and 499 mV for OER, corresponding with low Tafel slopes of 114.3 and 79.5 mV dec-1, respectively. Density functional theory (DFT) calculations demonstrate that modulation active sites with appropriate electronic properties facilitates the interaction between the catalyst surface and intermediates, especially for the adsorption of absorbed H* and *OOH intermediates, resulting in optimized energy barrier for HER and OER. The 3D nanoarray structure, with a large specific surface area and abundant ion channels, can enrich the electroactive sites and enhance mass transmission. This work provides a novel strategies and insights for the design of robust non-precious metal catalysts.

Introduction: Non-noble metal, including its hydroxides, chalcogenides, carbides, phosphides and nitrides have been reported as promising HER catalysts due to unique electronic properties and superior conductivity.[7-11] Among these materials, transition metal phosphides (TMPs) have attracted a great deal of attention owing to their unique electronic properties and great conductivity.[12-14] However, monometallic phosphides limited by their inherent properties, are difficult to achieve the desired catalytic performance for HER/OER. The multicomponent metal phosphides can realize the optimization of surface/interface electronic structures benefited from the synergistic effects to improve the HER and/or OER performance. Wang et al. have constructed Co3P/Ni2P heterojunction, achieving 10 mA cm−2 at an overpotential of 115 mV for HER.[15] The strong interactions at the heterojunction interface can optimize the electronic environment of catalytic surface, leading to suitable H+ adsorption and hydrogen formation kinetics. Wang and co-workers reported the phase-controlled synthesis of iron/nickel phosphides nanocrystals “armored” with porous P-doped carbon, Using (NixFe1-x)2P@PC/PG as cathode and anode catalysts, it needs only low cell voltage (1.45 V) to reach 10 mA cm−2 current.[16] Those outstanding works confirms that the fabrication of TMPs with multi-metal-component can mediate the electronic structure of TMPs, thereby provide more appropriate activity sites for HER/OER. A series of strategies have been explored, such as doping, bimetallic phosphides synthesis, heterostructure constructure, and interfacial engineering et. al. [15] While, the systemically modulation of electronic properties of active sites of bimetallic phosphating alloys in combination with its compositions and micromorphology is still challenging. The catalytic performance of the catalyst is not only related to the intrinsic properties of active sites, but also connected with the morphology.

Question 2. The authors have reported that “the Ni 2p spectrum peaks at 857.0 and 874.6 eV can be attributed to Niδ+”. The authors should define the δ, 3+ or 2+?

Response: Thank you for your rigorous review. The peaks at 857.0 and 874.6 eV can be indixed to the nickel with mixed valence state of bivalence and tervalence, which are related to Ni-P and oxidized nickel of Ni-POx. δ is defined as a value greater than two less than three. The above discussion has been supplemented in the manuscript as follows: The high-resolution Ni 2p spectrum of Ni2P NA/NF exhibits characteristic peaks located at 853.4 and 869.7 eV, assigning to Ni0. The peaks at 857.0 and 874.6 eV can be attributed to Niδ+ (2<δ<3), which are related to Ni-P and oxidized nickel of Ni-POx (Figure 4c).

Question 3. The authors should cite the following manuscript (J. Phys. Chem. C 2021, 125, 38, 20972–20979), reporting similar systems to the current manuscript. 

Response: Thank you for your constructive suggestions. These literatures are very helpful in improving the quality of our manuscript and have been cited as Ref. 28.

[28] Ramadoss, M.; Chen, Y.; Chen, X.; Su, Z.; Karpuraranjith, M.; Yang, D.; Muralidharan, K., Iron-Modulated Three-Dimensional CoNiP Vertical Nanoarrays: An Exploratory Binder-Free Bifunctional Electrocatalyst for Efficient Overall Water Splitting. J. Phy. Chem. C 2021, 125, 20972-20979.

Question 4. In the manuscript, the XPS P p region has not been explained in detail. The authors should clarify the state of P (doped and formed compound) in the nanoarrays. Additionally, we observe a shift in P p positions. Can the authors add missing discussion on this point? 

Response: Thank you for your constructive suggestions. The P 2p region has been analysed in detail and the phenomenon of shift has also been explained and supplemented in manuscript as follows: For P 2p spectra, the peaks at 129.5 and 130.4 eV are corresponded to P 2p3/2 and P 2p1/2 in metal phosphides. The peak at 134.1 eV is indexed to the oxidized phosphorus species originated from the surface oxidation, corresponding to the Co/Ni oxide state mentioned above. The peaks of CoNiP NA/NF with different Co/Ni ratios located at higher binding energy than that of Co2P NA/NF and Ni2P NA/NF, indicating that more electrons are transferred to P atoms from bimetals (Figure 4d).[29-31]

The state of P in the nanoarrays can be insured by the high-resolution of Co 2p/Ni 2p spectrum.

Figure 4b shows the XPS fine spectra of Co 2p, the peaks of Co2P NA/NF appearing at 778.6 (2p3/2) and 793.5 (2p1/2) eV can be identified as Co2+, and the peaks appearing at 782.3 (2p3/2) and 798.7 (2p1/2) eV can be identified as Co3+. These peaks originate from the oxidized cobalt of Co-POx and Co-P bonding, respectively.[27, 28] The high-resolution Ni 2p spectrum of Ni2P NA/NF exhibits characteristic peaks located at 853.4 and 869.7 eV, assigning to Ni0. The peaks at 857.0 and 874.6 eV can be attributed to Niδ+ (2<δ<3), which are related to Ni-P and oxidized nickel of Ni-POx (Figure 4c).[29] The above discussion proved that Ni and P are bonded, thus forming a compound.

Question 5: According to the SEM images presented in Figure 2, the type and the amount of transition metal additives significantly affect the resulting morphology. Can the authors comment on the mechanism causing different morphologies by indicating/citing related works?

Response:Thanks for your constructive advise. We summarized the literatures on cobalt phosphide and nickel phosphide nanoarrays. As show in figure R1a-b, nickel phosphide usually presents the morphology of nanowire arrays.[R2, R3] As for cobalt phosphide, it exhibits the  typical morphology of nanosheet arrays (figure R1c-d).[R4, R5] Unfortunately, the driving factors for the formation of diffirent morphology based on different metal precursors are rarely studied, and it needs to be further investigated.

Figure R1. (a, b)The reported SEM images of Co2P and (c, d) Ni2P

[R2] Xie, L.; Zhang, R.; Cui, L.; Liu; D.; Hao, S.; Ma, Y.; Sun, X. High-performance electrolytic oxygen evolution in neutral media catalyzed by a cobalt phosphate nanoarray. Angew. Chem. Inter. Edit. 2017, 56, 1064-1068.

[R3] Yang, Z.; Pan, Q.; Wu, Z.; Xiang, W.; Fu, F.; Wang, Y.; Guo, X. Self-supported cobalt phosphate nanoarray with pseudocapacitive behavior: An efficient 3D anode material for sodium-ion batteries. J. Alloy Compd., 2020, 848, 156285.

[R4] Chang, J.; Li, K.; Wu, Z.; Ge, J.; Liu, C.; Xing, W., Sulfur-doped nickel phosphide nanoplates arrays: a monolithic electrocatalyst for efficient hydrogen evolution reactions. ACS App. Mater. Inter. 2018, 10, 26303-26311.

[R5] Wang, P.; Pu, Z.; Li, Y.; Wu, L.; Tu, Z.; Jiang, M.; Mu, S., Iron-doped nickel phosphide nanosheet arrays: an efficient bifunctional electrocatalyst for water splitting. ACS appl. Mater. Inter. 2017, 9, 26001-26007.

Question 6. The authors should consider adding a comparison of their electrode with the published works as a table or a brief discussion. 

Response: Thanks for your constructive suggestion. The comparison tables for HER and overall water splitting performance have been supplimented to supplimentary information and discussed in the manuscript as follows: (1) CoNiP NA/NF-2 presents remarkable improved HER activity with a small overpotential of 162 mV at η10, smaller than Co2P/NF (177 mV) and Ni2P/NF (182 mV), demonstrating the increased activity of binary metal phosphide as compared to mono-metal phosphide. It is ahead of current reporting (Table S1). (2) As shown in Figure 7a, When CoNiP NA/NF-2 serves as both cathode and anode, it needs only 1.613 V to reach the current density of 10 mA cm-2. It is comparable to the reported TMPs-based bifunctional electrocatalysts (Table S2).

Table S1. The comparison of HER performance of the samples in 1 M KOH solution.

Catalysts

Current density (mA/cm2)

Overpotential

Ref

CoP-MNA/NF

10

189

Ni

10

400

ACS Catal. 2013, 3, 166

CoP/CC

10

209

J. Am. Chem. Soc. 2014, 136, 7587

Co2P nanorods

10

171

Nano Energy 2014, 9, 373

Fe-CoP

10

190

Adv. Sci., 2018, 5, 1800949.

CoFePBA@CoP

10

171

Small Methods, 2021, 5.

CoMnP/Ni2P

10

209

J. Mater. Chem. A, 2021, 9,22129-22139.

Ni2P-Fe2P

10

261

J. Mater. Chem. A, 2021, 9,

22129-22139.

CoNiP NA/NF

10

162

This works

Table S2. Comparison of cell potentials at a current density of 10 mA cm−2 for TMPs-based bifunctional electrocatalysts toward overall water splitting in 1.0 M KOH

Catalysts

Cell potential (V)

Ref

Ni5P4 film

1.7

Angew. Chem. Int. Ed. 2015, 54, 12361–12365

CoNiP/rGO

1.59

Adv. Funct. Mater. 2016, 26, 6785 6796

CoNiP

1.57

Adv. Funct. Mater. 2016, 26, 7644 7651

NiFeP

1.65

Adv. Energy Mater. 2017, 7, 1700107

CoNiP

1.64

Adv. Mater. Interfaces 2016, 3, 1500454

NiFeP

1.67

ACS Appl. Mater. Interfaces 2017, 9, 26134 26142

Ni1.5Fe0.5P

1.59

Nano Energy 2017, 34, 472 480

Ni1−xCoxP

1.59

Nanoscale 2016, 8, 19129 19138

CoNiP NA/NF

1.613

This works

Question 7. We observe a shift in the XRD peak positions for Co:Ni=1:2 and 0:3. Can the authors comment on this?

Response: Thanks for your constructive question. The peak position of CoNi-hydroxides NA/NF (Co:Ni=1:2) shifted to lower angle in compared with that of Ni-hydroxides NA/NF (Co:Ni=0:3). This indicates the increase of lattice constant, which is attributed to the incorporation of object atom (Ni) into Ni-hydroxides. And as the Co/Ni ratio increases, the trend of peak shifts increases, further demonstrating the above conclusions. This phenomenon proves the formation of CoNi-hydroxides alloys. The above discussion has been supplemented in manuscript.

R

Figure S1. XRD patterns of (a) CoNi-hydroxides NA/NF with different Co/Ni ratio.

Question 8. In Figure 5e and 6e, the authors present the EIS data, and equivalent circuit as an inset and briefly discuss the Rct without presenting fitted results. The authors should add the missing information. 

Response: Thanks for your constructive suggestion. The detailed fitted results and further analysis for EIS measurements have been supplimented to the manuscript as follows: (1) The electrochemical impedance spectroscopy (EIS) was employed to further explore the charge-transfer mechanism (Figure 5e). The CoNiP NA/NF-2 exhibits the smallest Rct values of 4.97 Ω than Co2P/NF (7.42 Ω), CoNiP NA/NF-1 (7.27 Ω), CoNiP/NF-3 (7.77 Ω), and Ni2P/NF (12.79 Ω), implying its faster charge transfer process. (2) The CoNiP NA/NF-2 also exhibits the smallest semicircle with Rct values of 4.97 Ω, demonstrating its superior charge transmission and lower charge-transfer resistance (Figure 6e).

Question 9. The authors have mentioned high specific surface area in the abstract and ECSA in the experimental section, however, the surface area measurements have not been given. Can the authors add the variation of BET surface area and electroactive surface area for each sample?

Response: Thanks for your constructive question. Figure S3a-b presents the nitrogen adsorption-desorption isotherm and the corresponding pore size distribution of CoNiP NA/NF-2. The nitrogen adsorption-desorption isotherm presents typical shape of the hysteresis loop, manifesting that abundant micropores and mesopores are existed in CoNiP NA/NF-2. According to the calculated BET surface area, the CoNiP NA/NF-2 possesses large surface area of 145.7 m2 g-1, corresponding with a wide mesopore size distribution in the range of 9~45 nm. The 3D cobalt-nickel phosphide nanoarray in situ grown on Ni foam catalyst has provided larger surface area and porous structure, thus a more favorable mass transfer processes can be carried out, thereby manifesting enhanced catalytic performance.

Figure S3 (a) Nitrogen adsorption-desorption isotherms and (b) corresponding pore size distribution of CoNiP NA/NF-2

Question 10. TEM elemental mapping shown in Figure 3d, nanoarrays are richer in P compared to Ni, Co or O. Can some of the P be saturated/adsorbed on the surface, and if there is any influence of excess P at the surface on electrocatalytic performance?

Response: Thank you for your constructive question.  According to TEM elemental mapping, P species presents obviously stronger signal in compared to Ni, Co or O species. Some P species may adsorbed on the surface of CoNi-NA/NF.  The “ensemble” P  may have a positive effect on catalytic performance. Accordign to the previous reports, the P atom on the surface of transition metal phosphides could not only be a “proton trap” to adsorb andcapture protons, but also be a “booster” to bring a high activation energy to the reaction, which makes hydrogen molecules desorption from the catalytic interface.

[R5] Chen, D.; Pu, Z.; Lu, R.; Ji, P.; Wang, P.; Zhu, J.; Mu, S. Ultralow Ru loading transition metal phosphides as high-efficient bifunctional electrocatalyst for a solar-to-hydrogen generation system. Adv. Energ. Mater. 2020, 10, 2000814.

[R6] Liu, P.; Rodriguez, J. A. Catalysts for hydrogen evolution from the [NiFe] hydrogenase to the Ni2P (001) surface: the importance of ensemble effect. J. Am. Chem. Soc. 2005, 127, 14871-14878.

Question 11. In the experimental section “Cyclic voltammetry (CV) measurements scanning were used to calculate the double-layer capacitance (Cdl) at non-Faradaic potential regions with...” and in the discussion part, some Cdl results are presented but how the authors did the calculations not given. For readers from diverse backgrounds, the authors should add the missing equation and cyclic voltammetry results.

 Response: Thank you for your constructive suggestions. cyclic voltammetry curves for HER and OER have been supplimented to Figure S3-4. The calculation method was also supplimented to the manuscript as follows: Cyclic voltammetry (CV) measurements scanning was measured to calculate the double layer capacitance (Cdl) at non-faradaic potential regions with different scan rates including 15, 20, 25, 30, 35 and 40 mV s-1. Then plotting the curve with regards to the current density and scan rate at a certain potential, and the slope of the curves is the value of Cdl.

Figure S4. (a-f) CV curves measured for HER with scan rate from 15 to 40 mV s‒1 for various samples.

Figure S5. (a-f) CV curves measured for OER with scan rate from 15 to 40 mV s‒1 for various samples.

Question 12. Minor mistakes

Line 102 “Co/Ni ratio and Co-hydroxides NA/NF, as well as Co-hydroxides NA/NF.....” and line 105 “Co-hydroxides NA/NF and Co-hydroxides NA/NF....”

Line 76 “Such comparisons of different metal phosphide HER catalysts with the same.”

Line 78 “....an critical role in 78 its catalytic performance.”

Response: Thank you for your careful review. (1) “Co/Ni ratio and Co-hydroxides NA/NF, as well as Co-hydroxides NA/NF.....” in line 102 and “Co-hydroxides NA/NF and Co-hydroxides NA/NF....” in line 105  have been modified as “SEM was firstly employed to illustrate the morphologies and microstructures of Ni foam and the as-synthesized Co-hydroxide NA/NF, Ni-hydroxide NA/NF, as well as CoNi-hydroxides NA/NF with different Co/Ni ratio.” and “Co-hydroxide NA/NF and Ni-hydroxides NA/NF” (2) “Such comparisons of different metal phosphide HER catalysts with the same.” in line 76 has been deleted. (3) “....an critical role in 78 its catalytic performance.”in line 78 has been modified as “ The monolithic integrated nanoarray catalysts carry a more stable structure, larger surface area for more active site exposure, and porous structure, thus a more favorable mass transfer processes can be carried out, thereby manifesting enhanced catalytic performance.”

we have checked the entire manuscript to avoid such errors before resubmitting.

Reviewer 2 Report

In the present manuscript, the authors assess the HER and OER performance of CoNiP nanoarrays grown on Ni foam. The Co, Ni or NiCo nanoarrays were prepared on Ni foam using hydrothermal method and then the samples were treated in P to form CoNiP nanoarrays. The authors have reported improved activity in HER and OER regions of linear sweep voltammetry. The authors have used theoretical and experimental approaches to investigate the systems and the work is interesting to the readers. However, the novelty of the work is not well-defined, crucial experimental data and discussions (listed below) are missing in the manuscript. Therefore, the current version of the manuscript is not suitable for publication. 

-The motivation and novelty of the work have not been clearly defined in the abstract. The authors should emphasize these points in the abstract. 

-The authors have reported that “the Ni2p spectrum peaks at 857.0 and 874.6 eV can be attributed to Niδ+”. The authors should define the δ, 3+ or 2+?

-The authors should cite the following manuscript (J. Phys. Chem. C 2021, 125, 38, 20972–20979), reporting similar systems to the current manuscript. 

-In the manuscript, the XPS P2p region has not been explained in detail. The authors should clarify the state of P (doped and formed compound) in the nanoarrays. Additionally, we observe a shift in P2p positions. Can the authors add missing discussion on this point? 

-According to the SEM images presented in Figure 2, the type and the amount of transition metal additives significantly affect the resulting morphology. Can the authors comment on the mechanism causing different morphologies by indicating/citing related works?

-The authors should consider adding a comparison of their electrode with the published works as a table or a brief discussion. 

-We observe a shift in the XRD peak positions for Co:Ni=1:2 and 0:3. Can the authors comment on this?

-In Figure 5e and 6e, the authors present the EIS data, and equivalent circuit as an inset and briefly discuss the Rct without presenting fitted results. The authors should add the missing information. 

-The authors have mentioned high specific surface area in the abstract and ECSA in the experimental section, however, the surface area measurements have not been given. Can the authors add the variation of BET surface area and electroactive surface area for each sample?

-TEM elemental mapping shown in Figure 3d, nanoarrays are richer in P compared to Ni, Co or O. Can some of the P be saturated/adsorbed on the surface, and if there is any influence of excess P at the surface on electrocatalytic performance?

-In the experimental section “Cyclic voltammetry (CV) measurements scanning were used to calculate the double-layer capacitance (Cdl) at non-Faradaic potential regions with...” and in the discussion part, some Cdl results are presented but how the authors did the calculations not given. For readers from diverse backgrounds, the authors should add the missing equation and cyclic voltammetry results.

Minor mistakes

-Line 102 “Co/Ni ratio and Co-hydroxides NA/NF, as well as Co-hydroxides NA/NF.....” and line 105 “Co-hydroxides NA/NF and Co-hydroxides NA/NF....”

-Line 76 “Such comparisons of different metal phosphide HER catalysts with the same.”

-Line 78 “....an critical role in 78 its catalytic performance.”

Author Response

Dear respected editor and reviewers,

Thank you for your letter to inform us your welcome decision and the reviewer’s positive comments on our manuscript (ID: ijms-1680621). According to the reviewer’s comments and your suggestions, we have carefully revised our manuscript. Now we are resubmitting the revised version with a detailed point-to-point response to the reviewer’s comments, and list of revisions made. We hope that the revised manuscript is suitable for publication in your journal- International Journal of Molecular Sciences. If there are any questions about the revision, please feel free to contact me at any time. We thank the reviewers for their correct comments and look forward to hearing good news from you regarding our revised manuscript.

Best wishes!

Sincerely yours,

Prof. Zhikun Peng

Reviewer: 2

Comments:

In the present manuscript, the authors assess the HER and OER performance of CoNiP nanoarrays grown on Ni foam. The Co, Ni or NiCo nanoarrays were prepared on Ni foam using hydrothermal method and then the samples were treated in P to form CoNiP nanoarrays. The authors have reported improved activity in HER and OER regions of linear sweep voltammetry. The authors have used theoretical and experimental approaches to investigate the systems and the work is interesting to the readers. However, the novelty of the work is not well-defined, crucial experimental data and discussions (listed below) are missing in the manuscript. Therefore, the current version of the manuscript is not suitable for publication. 

Question 1. The motivation and novelty of the work have not been clearly defined in the abstract. The authors should emphasize these points in the abstract. (abstract,introduction)

Response: Thank you for your constructive suggestions. The motivations and innovations of the manuscript  have been redistilled and highlighted in the abstract and introduction as follows:Abstract: Careful designing of nanostructures and multi-compositions of non-noble metal based electrocatalysts for highly efficient electrocatalytic hydrogen and oxygen evolution reaction (HER and OER) is of great significance to realize sustainable hydrogen release. Herein, bifunctional electrocatalysts of the three-dimensional (3D) cobalt-nickel phosphide nanoarray in situ grown on nickel foams (CoNiP NA/NF) were synthesized through a facile hydrothermal method followed by phosphorization. Due to the unique self-template nanoarray structure and tunable multicomponent system, the CoNiP NA/NF samples presents exceptional activity and durability for HER and OER. The optimized sample of CoNiP NA/NF-2 afforded a current density of 10 mA cm-2 at a low overpotential of 162 mV for HER and 499 mV for OER, corresponding with low Tafel slopes of 114.3 and 79.5 mV dec-1, respectively. Density functional theory (DFT) calculations demonstrate that modulation active sites with appropriate electronic properties facilitates the interaction between the catalyst surface and intermediates, especially for the adsorption of absorbed H* and *OOH intermediates, resulting in optimized energy barrier for HER and OER. The 3D nanoarray structure, with a large specific surface area and abundant ion channels, can enrich the electroactive sites and enhance mass transmission. This work provides a novel strategies and insights for the design of robust non-precious metal catalysts.

Introduction: Non-noble metal, including its hydroxides, chalcogenides, carbides, phosphides and nitrides have been reported as promising HER catalysts due to unique electronic properties and superior conductivity.[7-11] Among these materials, transition metal phosphides (TMPs) have attracted a great deal of attention owing to their unique electronic properties and great conductivity.[12-14] However, monometallic phosphides limited by their inherent properties, are difficult to achieve the desired catalytic performance for HER/OER. The multicomponent metal phosphides can realize the optimization of surface/interface electronic structures benefited from the synergistic effects to improve the HER and/or OER performance. Wang et al. have constructed Co3P/Ni2P heterojunction, achieving 10 mA cm−2 at an overpotential of 115 mV for HER.[15] The strong interactions at the heterojunction interface can optimize the electronic environment of catalytic surface, leading to suitable H+ adsorption and hydrogen formation kinetics. Wang and co-workers reported the phase-controlled synthesis of iron/nickel phosphides nanocrystals “armored” with porous P-doped carbon, Using (NixFe1-x)2P@PC/PG as cathode and anode catalysts, it needs only low cell voltage (1.45 V) to reach 10 mA cm−2 current.[16] Those outstanding works confirms that the fabrication of TMPs with multi-metal-component can mediate the electronic structure of TMPs, thereby provide more appropriate activity sites for HER/OER. A series of strategies have been explored, such as doping, bimetallic phosphides synthesis, heterostructure constructure, and interfacial engineering et. al. [15] While, the systemically modulation of electronic properties of active sites of bimetallic phosphating alloys in combination with its compositions and micromorphology is still challenging. The catalytic performance of the catalyst is not only related to the intrinsic properties of active sites, but also connected with the morphology.

Question 2. The authors have reported that “the Ni 2p spectrum peaks at 857.0 and 874.6 eV can be attributed to Niδ+”. The authors should define the δ, 3+ or 2+?

Response: Thank you for your rigorous review. The peaks at 857.0 and 874.6 eV can be indixed to the nickel with mixed valence state of bivalence and tervalence, which are related to Ni-P and oxidized nickel of Ni-POx. δ is defined as a value greater than two less than three. The above discussion has been supplemented in the manuscript as follows: The high-resolution Ni 2p spectrum of Ni2P NA/NF exhibits characteristic peaks located at 853.4 and 869.7 eV, assigning to Ni0. The peaks at 857.0 and 874.6 eV can be attributed to Niδ+ (2<δ<3), which are related to Ni-P and oxidized nickel of Ni-POx (Figure 4c).

Question 3. The authors should cite the following manuscript (J. Phys. Chem. C 2021, 125, 38, 20972–20979), reporting similar systems to the current manuscript. 

Response: Thank you for your constructive suggestions. These literatures are very helpful in improving the quality of our manuscript and have been cited as Ref. 28.

[28] Ramadoss, M.; Chen, Y.; Chen, X.; Su, Z.; Karpuraranjith, M.; Yang, D.; Muralidharan, K., Iron-Modulated Three-Dimensional CoNiP Vertical Nanoarrays: An Exploratory Binder-Free Bifunctional Electrocatalyst for Efficient Overall Water Splitting. J. Phy. Chem. C 2021, 125, 20972-20979.

Question 4. In the manuscript, the XPS P p region has not been explained in detail. The authors should clarify the state of P (doped and formed compound) in the nanoarrays. Additionally, we observe a shift in P p positions. Can the authors add missing discussion on this point? 

Response: Thank you for your constructive suggestions. The P 2p region has been analysed in detail and the phenomenon of shift has also been explained and supplemented in manuscript as follows: For P 2p spectra, the peaks at 129.5 and 130.4 eV are corresponded to P 2p3/2 and P 2p1/2 in metal phosphides. The peak at 134.1 eV is indexed to the oxidized phosphorus species originated from the surface oxidation, corresponding to the Co/Ni oxide state mentioned above. The peaks of CoNiP NA/NF with different Co/Ni ratios located at higher binding energy than that of Co2P NA/NF and Ni2P NA/NF, indicating that more electrons are transferred to P atoms from bimetals (Figure 4d).[29-31]

The state of P in the nanoarrays can be insured by the high-resolution of Co 2p/Ni 2p spectrum.

Figure 4b shows the XPS fine spectra of Co 2p, the peaks of Co2P NA/NF appearing at 778.6 (2p3/2) and 793.5 (2p1/2) eV can be identified as Co2+, and the peaks appearing at 782.3 (2p3/2) and 798.7 (2p1/2) eV can be identified as Co3+. These peaks originate from the oxidized cobalt of Co-POx and Co-P bonding, respectively.[27, 28] The high-resolution Ni 2p spectrum of Ni2P NA/NF exhibits characteristic peaks located at 853.4 and 869.7 eV, assigning to Ni0. The peaks at 857.0 and 874.6 eV can be attributed to Niδ+ (2<δ<3), which are related to Ni-P and oxidized nickel of Ni-POx (Figure 4c).[29] The above discussion proved that Ni and P are bonded, thus forming a compound.

Question 5: According to the SEM images presented in Figure 2, the type and the amount of transition metal additives significantly affect the resulting morphology. Can the authors comment on the mechanism causing different morphologies by indicating/citing related works?

Response:Thanks for your constructive advise. We summarized the literatures on cobalt phosphide and nickel phosphide nanoarrays. As show in figure R1a-b, nickel phosphide usually presents the morphology of nanowire arrays.[R2, R3] As for cobalt phosphide, it exhibits the  typical morphology of nanosheet arrays (figure R1c-d).[R4, R5] Unfortunately, the driving factors for the formation of diffirent morphology based on different metal precursors are rarely studied, and it needs to be further investigated.

Figure R1. (a, b)The reported SEM images of Co2P and (c, d) Ni2P

[R2] Xie, L.; Zhang, R.; Cui, L.; Liu; D.; Hao, S.; Ma, Y.; Sun, X. High-performance electrolytic oxygen evolution in neutral media catalyzed by a cobalt phosphate nanoarray. Angew. Chem. Inter. Edit. 2017, 56, 1064-1068.

[R3] Yang, Z.; Pan, Q.; Wu, Z.; Xiang, W.; Fu, F.; Wang, Y.; Guo, X. Self-supported cobalt phosphate nanoarray with pseudocapacitive behavior: An efficient 3D anode material for sodium-ion batteries. J. Alloy Compd., 2020, 848, 156285.

[R4] Chang, J.; Li, K.; Wu, Z.; Ge, J.; Liu, C.; Xing, W., Sulfur-doped nickel phosphide nanoplates arrays: a monolithic electrocatalyst for efficient hydrogen evolution reactions. ACS App. Mater. Inter. 2018, 10, 26303-26311.

[R5] Wang, P.; Pu, Z.; Li, Y.; Wu, L.; Tu, Z.; Jiang, M.; Mu, S., Iron-doped nickel phosphide nanosheet arrays: an efficient bifunctional electrocatalyst for water splitting. ACS appl. Mater. Inter. 2017, 9, 26001-26007.

Question 6. The authors should consider adding a comparison of their electrode with the published works as a table or a brief discussion. 

Response: Thanks for your constructive suggestion. The comparison tables for HER and overall water splitting performance have been supplimented to supplimentary information and discussed in the manuscript as follows: (1) CoNiP NA/NF-2 presents remarkable improved HER activity with a small overpotential of 162 mV at η10, smaller than Co2P/NF (177 mV) and Ni2P/NF (182 mV), demonstrating the increased activity of binary metal phosphide as compared to mono-metal phosphide. It is ahead of current reporting (Table S1). (2) As shown in Figure 7a, When CoNiP NA/NF-2 serves as both cathode and anode, it needs only 1.613 V to reach the current density of 10 mA cm-2. It is comparable to the reported TMPs-based bifunctional electrocatalysts (Table S2).

Table S1. The comparison of HER performance of the samples in 1 M KOH solution.

Catalysts

Current density (mA/cm2)

Overpotential

Ref

CoP-MNA/NF

10

189

Ni

10

400

ACS Catal. 2013, 3, 166

CoP/CC

10

209

J. Am. Chem. Soc. 2014, 136, 7587

Co2P nanorods

10

171

Nano Energy 2014, 9, 373

Fe-CoP

10

190

Adv. Sci., 2018, 5, 1800949.

CoFePBA@CoP

10

171

Small Methods, 2021, 5.

CoMnP/Ni2P

10

209

J. Mater. Chem. A, 2021, 9,22129-22139.

Ni2P-Fe2P

10

261

J. Mater. Chem. A, 2021, 9,

22129-22139.

CoNiP NA/NF

10

162

This works

Table S2. Comparison of cell potentials at a current density of 10 mA cm−2 for TMPs-based bifunctional electrocatalysts toward overall water splitting in 1.0 M KOH

Catalysts

Cell potential (V)

Ref

Ni5P4 film

1.7

Angew. Chem. Int. Ed. 2015, 54, 12361–12365

CoNiP/rGO

1.59

Adv. Funct. Mater. 2016, 26, 6785 6796

CoNiP

1.57

Adv. Funct. Mater. 2016, 26, 7644 7651

NiFeP

1.65

Adv. Energy Mater. 2017, 7, 1700107

CoNiP

1.64

Adv. Mater. Interfaces 2016, 3, 1500454

NiFeP

1.67

ACS Appl. Mater. Interfaces 2017, 9, 26134 26142

Ni1.5Fe0.5P

1.59

Nano Energy 2017, 34, 472 480

Ni1−xCoxP

1.59

Nanoscale 2016, 8, 19129 19138

CoNiP NA/NF

1.613

This works

Question 7. We observe a shift in the XRD peak positions for Co:Ni=1:2 and 0:3. Can the authors comment on this?

Response: Thanks for your constructive question. The peak position of CoNi-hydroxides NA/NF (Co:Ni=1:2) shifted to lower angle in compared with that of Ni-hydroxides NA/NF (Co:Ni=0:3). This indicates the increase of lattice constant, which is attributed to the incorporation of object atom (Ni) into Ni-hydroxides. And as the Co/Ni ratio increases, the trend of peak shifts increases, further demonstrating the above conclusions. This phenomenon proves the formation of CoNi-hydroxides alloys. The above discussion has been supplemented in manuscript.

R

Figure S1. XRD patterns of (a) CoNi-hydroxides NA/NF with different Co/Ni ratio.

Question 8. In Figure 5e and 6e, the authors present the EIS data, and equivalent circuit as an inset and briefly discuss the Rct without presenting fitted results. The authors should add the missing information. 

Response: Thanks for your constructive suggestion. The detailed fitted results and further analysis for EIS measurements have been supplimented to the manuscript as follows: (1) The electrochemical impedance spectroscopy (EIS) was employed to further explore the charge-transfer mechanism (Figure 5e). The CoNiP NA/NF-2 exhibits the smallest Rct values of 4.97 Ω than Co2P/NF (7.42 Ω), CoNiP NA/NF-1 (7.27 Ω), CoNiP/NF-3 (7.77 Ω), and Ni2P/NF (12.79 Ω), implying its faster charge transfer process. (2) The CoNiP NA/NF-2 also exhibits the smallest semicircle with Rct values of 4.97 Ω, demonstrating its superior charge transmission and lower charge-transfer resistance (Figure 6e).

Question 9. The authors have mentioned high specific surface area in the abstract and ECSA in the experimental section, however, the surface area measurements have not been given. Can the authors add the variation of BET surface area and electroactive surface area for each sample?

Response: Thanks for your constructive question. Figure S3a-b presents the nitrogen adsorption-desorption isotherm and the corresponding pore size distribution of CoNiP NA/NF-2. The nitrogen adsorption-desorption isotherm presents typical shape of the hysteresis loop, manifesting that abundant micropores and mesopores are existed in CoNiP NA/NF-2. According to the calculated BET surface area, the CoNiP NA/NF-2 possesses large surface area of 145.7 m2 g-1, corresponding with a wide mesopore size distribution in the range of 9~45 nm. The 3D cobalt-nickel phosphide nanoarray in situ grown on Ni foam catalyst has provided larger surface area and porous structure, thus a more favorable mass transfer processes can be carried out, thereby manifesting enhanced catalytic performance.

Figure S3 (a) Nitrogen adsorption-desorption isotherms and (b) corresponding pore size distribution of CoNiP NA/NF-2

Question 10. TEM elemental mapping shown in Figure 3d, nanoarrays are richer in P compared to Ni, Co or O. Can some of the P be saturated/adsorbed on the surface, and if there is any influence of excess P at the surface on electrocatalytic performance?

Response: Thank you for your constructive question.  According to TEM elemental mapping, P species presents obviously stronger signal in compared to Ni, Co or O species. Some P species may adsorbed on the surface of CoNi-NA/NF.  The “ensemble” P  may have a positive effect on catalytic performance. Accordign to the previous reports, the P atom on the surface of transition metal phosphides could not only be a “proton trap” to adsorb andcapture protons, but also be a “booster” to bring a high activation energy to the reaction, which makes hydrogen molecules desorption from the catalytic interface.

[R5] Chen, D.; Pu, Z.; Lu, R.; Ji, P.; Wang, P.; Zhu, J.; Mu, S. Ultralow Ru loading transition metal phosphides as high-efficient bifunctional electrocatalyst for a solar-to-hydrogen generation system. Adv. Energ. Mater. 2020, 10, 2000814.

[R6] Liu, P.; Rodriguez, J. A. Catalysts for hydrogen evolution from the [NiFe] hydrogenase to the Ni2P (001) surface: the importance of ensemble effect. J. Am. Chem. Soc. 2005, 127, 14871-14878.

Question 11. In the experimental section “Cyclic voltammetry (CV) measurements scanning were used to calculate the double-layer capacitance (Cdl) at non-Faradaic potential regions with...” and in the discussion part, some Cdl results are presented but how the authors did the calculations not given. For readers from diverse backgrounds, the authors should add the missing equation and cyclic voltammetry results.

 Response: Thank you for your constructive suggestions. cyclic voltammetry curves for HER and OER have been supplimented to Figure S3-4. The calculation method was also supplimented to the manuscript as follows: Cyclic voltammetry (CV) measurements scanning was measured to calculate the double layer capacitance (Cdl) at non-faradaic potential regions with different scan rates including 15, 20, 25, 30, 35 and 40 mV s-1. Then plotting the curve with regards to the current density and scan rate at a certain potential, and the slope of the curves is the value of Cdl.

Figure S4. (a-f) CV curves measured for HER with scan rate from 15 to 40 mV s‒1 for various samples.

Figure S5. (a-f) CV curves measured for OER with scan rate from 15 to 40 mV s‒1 for various samples.

Question 12. Minor mistakes

Line 102 “Co/Ni ratio and Co-hydroxides NA/NF, as well as Co-hydroxides NA/NF.....” and line 105 “Co-hydroxides NA/NF and Co-hydroxides NA/NF....”

Line 76 “Such comparisons of different metal phosphide HER catalysts with the same.”

Line 78 “....an critical role in 78 its catalytic performance.”

Response: Thank you for your careful review. (1) “Co/Ni ratio and Co-hydroxides NA/NF, as well as Co-hydroxides NA/NF.....” in line 102 and “Co-hydroxides NA/NF and Co-hydroxides NA/NF....” in line 105  have been modified as “SEM was firstly employed to illustrate the morphologies and microstructures of Ni foam and the as-synthesized Co-hydroxide NA/NF, Ni-hydroxide NA/NF, as well as CoNi-hydroxides NA/NF with different Co/Ni ratio.” and “Co-hydroxide NA/NF and Ni-hydroxides NA/NF” (2) “Such comparisons of different metal phosphide HER catalysts with the same.” in line 76 has been deleted. (3) “....an critical role in 78 its catalytic performance.”in line 78 has been modified as “ The monolithic integrated nanoarray catalysts carry a more stable structure, larger surface area for more active site exposure, and porous structure, thus a more favorable mass transfer processes can be carried out, thereby manifesting enhanced catalytic performance.”

we have checked the entire manuscript to avoid such errors before resubmitting.

Round 2

Reviewer 1 Report

The authors have considered all the suggestions made and improved the manuscript accordingly. I can now support publication. 

Reviewer 2 Report

The authors have carefully addressed all comments and concerns raised by the reviewer. I would recommend this manuscript be accepted.